# Dietary-Derived Essential Nutrients and Amyotrophic Lateral Sclerosis: A Two-Sample Mendelian Randomization Study

**DOI:** 10.3390/nu14050920

**Published:** 2022-02-22

**Authors:** Kailin Xia, Yajun Wang, Linjing Zhang, Lu Tang, Gan Zhang, Tao Huang, Ninghao Huang, Dongsheng Fan

**Affiliations:** 1Department of Neurology, Peking University Third Hospital, Beijing 100191, China; kllook@pku.edu.cn (K.X.); bttre_wyj@sina.com (Y.W.); zhanglinjing@bjmu.edu.cn (L.Z.); tanglu@bjmu.edu.cn (L.T.); nmzhanggan@yahoo.com (G.Z.); 2Beijing Key Laboratory of Biomarker and Translational Research in Neurodegenerative Diseases, Beijing 100191, China; 3Key Laboratory for Neuroscience, National Health Commission/Ministry of Education, Peking University, Beijing 100191, China; 4Department of Epidemiology and Biostatistics, School of Public Health, Peking University, Beijing 100871, China; huangtao@bjmu.edu.cn (T.H.); ninghaohuang@bjmu.edu.cn (N.H.)

**Keywords:** amyotrophic lateral sclerosis, nutrition, Mendelian randomization, genes, risk factor

## Abstract

Previous studies have suggested a close but inconsistent relationship between essential nutrients and the risk of amyotrophic lateral sclerosis (ALS), and whether this association is causal remains unknown. We aimed to investigate the potential causal relation between essential nutrients (essential amino acids, essential fatty acids, essential minerals, and essential vitamins) and the risk of ALS using Mendelian randomization (MR) analysis. Large-scale European-based genome-wide association studies’ (GWASs) summary data related to ALS (assembling 27,205 ALS patients and 110,881 controls) and essential nutrient concentrations were separately obtained. MR analysis was performed using the inverse variance–weighted (IVW) method, and sensitivity analysis was conducted by the weighted median method, simple median method, MR–Egger method and MR–PRESSO method. We found a causal association between genetically predicted linoleic acid (LA) and the risk of ALS (OR: 1.066; 95% CI: 1.011–1.125; *p* = 0.019). An inverse association with ALS risk was noted for vitamin D (OR: 0.899; 95% CI: 0.819–0.987; *p* = 0.025) and for vitamin E (OR: 0.461; 95% CI: 0.340–0.626; *p* = 6.25 × 10^−7^). The sensitivity analyses illustrated similar trends. No causal effect was observed between essential amino acids and minerals on ALS. Our study profiled the effects of diet-derived circulating nutrients on the risk of ALS and demonstrated that vitamin D and vitamin E are protective against the risk of ALS, and LA is a suggested risk factor for ALS.

## 1. Introduction

Amyotrophic lateral sclerosis (ALS) is a fatal neurodegenerative disease associated with muscle weakness and atrophy, and patients usually die as a result of respiratory failure within 3–5 years of symptom onset [1,2]. No treatment is currently effective in reversing the progression of ALS, with the exception of riluzole, which probably prolongs median survival by 2–3 months [3]. The etiology of ALS is complex, largely unknown, and involves genetic and environmental factors [4]. Given the accumulated studies investigating risk factors for ALS, the implication of such findings has been a focus. Targeted lifestyle and dietary changes have been shown to be beneficial in the early stages of ALS [5,6].

Essential nutrients accessible from the diet include essential amino acids, essential fatty acids, vitamins, and minerals. Supplementing essential nutrients is considered a promising preventive intervention for many diseases due to its ability to maintain metabolic homeostasis and to mitigate oxidative stress [7,8,9]. With efforts made to delineate the influence of essential nutrients on ALS, different conclusions have been drawn. For instance, a large, pooled prospective study reported that long-term vitamin E supplementation was inversely associated with the risk of ALS [10], while null results were found in two other case–control studies [11,12]. Similarly, controversial results were observed for docosahexaenoic acid (DHA) [13,14], iron [15,16], vitamin A, and vitamin C [11,17]. This inconsistency is largely because most of this evidence came from observational studies, which are plagued by selection bias and unmeasured confounders. In addition, we previously found that most ALS patients showed a loss of appetite [18] and the presence of metabolic disorders [19], which may affect the absorption and consumption of dietary nutrients, making the true relationship between essential nutrients and ALS elusive. Randomized controlled trials (RCTs) are recognized to overcome the limitations of observational studies and provide the highest level of evidence [20]. However, no previous RCT has been conducted to identify the risk/protective effect of essential nutrients on ALS.

Mendelian randomization (MR) analysis is a genetic statistical approach to estimate the causal association between an exposure and an outcome using single nucleotide polymorphisms as instrumental variables (IVs), which mimic the random assignment to groups of RCTs by naturally allocating alleles and can lessen residual confounding [21].

In this study, we performed a two-sample MR approach to systematically decipher the causal association between nutrition status and ALS.

## 2. Materials and Methods

### 2.1. Data Collection and Selection of Instrumental Variables (IVs)

Essential amino acids, essential fatty acids, essential minerals, and essential vitamins were considered essential nutrients. We searched PubMed to extract identified exposure-related SNPs from the most recent and largest genome-wide association studies (GWASs) based on the European population.

SNPs related to amino acids were extracted from seven datasets, namely, isoleucine, leucine, lysine, methionine, phenylalanine, tryptophan, and valine [22,23]. Arachidonic acid (AA), alpha-linolenic acid (ALA), DHA, docosapentaenoic acid (DPA), dihomo-gamma-linolenic acid (DGLA), eicosapentaenoic acid (EPA), gamma linolenic acid (GLA), and linoleic acid (LA) [22,24,25] were included in the essential fatty acid category. For the essential minerals, GWASs for calcium [26], copper, iron [27], magnesium [28], phosphorus [29], selenium [30], and zinc [31] were searched separately. In addition, we sought out updated calcium GWAS data in a preprint paper based on the UK Biobank with a fivefold enlarged sample size [32], and these data were recorded as a supplement to calcium, named “Calcium (UKB)”. To comprehensively assess the effects of vitamins and provitamins on disease, we examined both the absolute levels of vitamins or provitamins in blood and the relative concentrations of corresponding circulating metabolites quantified by a high-throughput platform. For absolute vitamin or provitamin levels, absolute vitamin A (retinol) [33], absolute beta-carotene [34], absolute vitamin B (vitamin B-6 and vitamin B-12) [35,36], absolute vitamin C (ascorbic acid) [37], absolute vitamin D (25-hydroxyvitamin D (25OHD)) [38], and absolute vitamin E (alpha-tocopherol) [39] were included, and for the relative concentrations of corresponding metabolites, relative retinol [40], relative vitamin C (ascorbate) [23], and relative vitamin E (alpha-tocopherol and gamma-tocopherol) [23] were used.

We obtained large-scale European-based ALS GWAS summary data from a recently published study, which included 27,205 ALS patients and 110,881 controls [41]. All patients were diagnosed according to the (revised) El Escorial criteria by specialized neurologists.

We set strict selection criteria for eligible IVs: (1) independent SNPs (r^2^ = 0.01, KB = 5000) with genome-wide association levels (*p* < 5 × 10^−8^) for exposure events; for exposures without genome-wide association significant IVs, suggestive level genome-wide association significant (*p* < 1 × 10^−5^) IVs were employed as proxies; (2) nonrare SNPs (MAF ≥ 0.01); and (3) proxies in strong linkage disequilibrium (r^2^ > 0.8) were employed to replace SNPs that failed to provide corresponding information for ALS. Proxy loci not included in the ALS dataset were excluded. The LDlink online tool [42] was employed in searching proxies. In total, we derived 1–168 genetic instruments for 35 essential nutritional biomarkers. Due to methodology limitations, exposures with at least two IVs were ultimately included in the downstream analysis. Lysine, methionine, ALA, absolute lycopene, absolute beta-carotene, and absolute vitamin B-6 were removed.

To measure the phenotypic variation explained by each SNP, R^2^ = 2 × beta^2^ × (1 − EAF) × EAF/SD^2^, where EAF = effect allele frequency and beta = effect of each SNP on exposures [43]. We similarly calculated F statistics for individual SNPs to indicate the strength of IVs (F = R^2^ × (N − 2)/((1 − R^2^))) [44]. Detailed information on SNPs is displayed in Appendix A.

### 2.2. Mendelian Randomization

Aiming to ensure the validity of the IVs, heterogeneity and directional pleiotropy were tested using Cochran’s Q test and the MR–PRESSO method. For Cochran’s Q test, nonsignificant heterogeneity (*p* > 0.05) was indicated by analogical estimates for each IV. For the MR–PRESSO method [45], (1) the overall horizontal pleiotropy was tested with the “global test”; (2) the “MR–PRESSO outlier test” was adopted to remove possible outliers in IVs and to generate corrected results; and (3) the “MR–PRESSO distortion test” was used to determine whether there was distortion between pre- and post-correction results. Similarly, the MR–Egger regression intercept was calculated as an additional means to examine horizontal pleiotropy. When there was no obvious distance from the intercept to the origin, it was regarded as having no influence on pleiotropy.

In this MR framework, the multiplicative random effect inverse variance–weighted method (IVW) was utilized as the main approach to analyze causality, which regresses the effect of SNPs on the outcome by the effect of the SNP on the exposure [46]. To ensure the robustness of the IVW results, additional sensitivity analyses were performed, including the simple median method, weighted median method and MR–Egger method. The weighted median method enables satisfactory estimates when at least 50% of the IVs are valid instruments [47]. The MR–Egger method can model terms to detect and correct the bias caused by pleiotropy, obtaining a consistent estimate of the causal effect [48]. We also implemented the “leave-one-out” (LOO) method to assess the influence of an individual SNP on the overall causal findings by excluding each genetic variant in turn and recomputing the MR–IVW estimates. The MR–Steiger method was employed to determine whether the direction of causality was true. Our flow chart was visualized in Figure 1.

*p* values were adjusted according to the strict Bonferroni correction. *p* < 0.05/k was considered a significant level (where k is the number of exposures), and a *p* value between 0.05/k and 0.05 was suggestive of a positive association. All analyses were performed in R software version 4.1.0 by the “TwoSampleMR” package (version 0.5.6) [49] and the “MR–PRESSO” package (version 1.0) [45].

## 3. Results

The 29 dietary-derived potential risk factors for ALS that we evaluated involved five traits related to amino acids, seven to fatty acids, eight to minerals, and nine to vitamins, including 2–168 SNPs as genetic proxies for each of them. All associations analyzed by IVW random-effects models are shown in Table 1.

In our restricted analysis of amino acid profiles and ALS risk, no association was noted between them (Table 1 and Table 2, Figure 2, and Appendix A). However, the IVW estimated for isoleucine and tryptophan may be vulnerable to the heterogeneity of SNPs (isoleucine: Cochran’s Q test *p* value = 0.037; tryptophan: Cochran’s Q test *p* value = 0). Due to the limited number of IVs in isoleucine, we cannot conduct further analysis. After removing the tryptophan outliers, we obtained analyzable results (*p*-outlier value = 0.893).

With regard to the effect of fatty acids on ALS, the results are summarized in Figure 3. We found a suggestive significant association between LA and the risk of ALS, which was elevated by 6.6% for a one standard deviation unit increase in the LA concentration (IVW-OR: 1.066, 95% CI: 1.011–1.125, *p* value = 0.019). The robustness of this result was supported by the MR–Egger approach (OR = 1.162, 95% CI: 1.029–1.312, *p* value = 0.028), and estimates using the weighted median method (OR = 1.049, 95% CI: 0.976–1.128, *p* value = 0.196) and simple median method (OR = 1.047, 95% CI: 0.972–1.128, *p* value = 0.226) also showed similar trends. No heterogeneity was detected in the individual SNPs associated with LA and the risk of ALS (Cochran’s Q test *p* value = 0.253). No indication of horizontal pleiotropy was found based on the MR–Egger intercept test (intercept = −0.014, *p* value = 0.146), and the MR–PRESSO analysis showed no outlying SNPs (Table 3). Scatter plots indicated the estimated effect of LA on ALS by each SNP (Figure 4A). In addition, we noted a positive significant association between DGLA and the risk of ALS (OR: 1.031, 95% CI: 1.017–1.045, *p* value = 1.70 × 10^−5^) without the influence of heterogeneity. Due to only two proxy SNPs being available for DGLA, sensitivity analyses could not be performed. The estimated effect of DGLA on ALS by each SNP is displayed in Figure 4B. Genetically predicted AA, DHA, DPA, EPA, and GLA were not associated with the risk of ALS (Table 1 and Table 2 and Appendix A).

We observed no causal relationship between circulating essential minerals and the risk of ALS (Table 1 and Table 2, Figure 5, and Appendix A). The MR–Egger regression did not identify evidence of horizontal pleiotropy for any of the mineral-related traits. Heterogeneity in the SNPs for iron was inferred (Cochran’s Q test *p* value = 0.034). However, because of the removal of SNPs with a null association with the overall iron status, this calculation was only based on three SNPs.

When information about IVs related to circulating vitamins was considered, the results generated by IVW analysis indicated that absolute 25OHD and relative alpha-tocopherol had strongly protective effects on ALS risk (absolute 25OHD, OR: 0.899, 95% CI: 0.819–0.987, *p* value = 0.025; and relative alpha-tocopherol, OR: 0.461; 95% CI: 0.340–0.626; *p* value = 6.25 × 10^−7^) (Table 1 and Table 3 and Figure 6). Both associations were broadly concordant in various sensitivity analyses. The results of the MR–Egger method (OR = 0.876, 95% CI: 0.774–0.993, *p* value = 0.042) and weighted median method (OR = 0.901, 95% CI: 0.800–1.014, *p* value = 0.084) underpinned the causal effect of 25OHD on ALS, while the estimates of the simple median method (OR = 0.617, 95% CI: 0.355–1.072, *p* value = 0.087) and weighted median method (OR  = 0.536, 95% CI: 0.324–0.886, *p* value = 0.015) were aligned with the significant relationship between relative alpha-tocopherol. No heterogeneity was detected using Cochran’s Q test separately (absolute 25OHD: *p* value = 0.110; relative alpha-tocopherol: *p* value = 0.758). The MR–Egger intercept test showed no evidence of directional pleiotropy (absolute 25OHD: intercept = 0.001, *p* value = 0.548; relative alpha-tocopherol: intercept = −0.065, *p* value = 0.643), and no outlying SNP was found. The scatter plots indicated the SNP-based causal estimates of absolute 25OHD on ALS (Figure 4C) and relative alpha-tocopherol on ALS (Figure 4D). The LOO analysis showed that SNPs rs577185477 and rs11723621 at a known risk locus for absolute 25OHD influenced the causal estimate for ALS (Appendix A). No effect of individual relative alpha-tocopherol-related SNPs on ALS risk was observed (Appendix A). A null effect of genetically predicted increases in circulating levels of absolute retinol, absolute vitamin B-12, absolute ascorbate, absolute alpha-tocopherol, relative retinol, relative ascorbate, and relative gamma-tocopherol on the risk of ALS was found (Table 1 and Table 2, Figure 6, and Appendix A).

## 4. Discussion

In the present MR study, we classified and profiled the potential causal role of essential dietary nutritiona6l modifiers on ALS risk using GWAS summary statistics. Higher genetically predicted levels of 25OHD and relative alpha-tocopherol were calculated to be associated with a reduced risk of ALS, and increased LA levels showed a suggestive positive association with ALS risk. Our findings implied that the abnormal levels of these nutrients were not only a symptom of ALS but also independent risk/protective factors for ALS. Among the essential amino acids and minerals, no evidence supports that additional supplementation has an influence on the risk of ALS.

Essential fatty acids are important components of cholesterol and phospholipids and are involved in the maintenance of membrane fluidity and beta-oxidation [50]. Our results showed that LA was a major risk factor for the effects of essential fatty acids on ALS, preliminarily suggesting that dietary supplementation with LA is not beneficial or is even harmful to ALS. As secondary evidence, we found that DGLA, which is derived from LA and is capable of anti-inflammatory and antiproliferative effects, had a strong risk effect on ALS [51,52]. LA is an n-6 polyunsaturated fatty acid. Because of the deficiency of desaturases that insert double bonds beyond C9 in humans, LA is accessible only from diet [53]. Although there is no direct clinical evidence for the neurotoxic effect of LA, a longitudinal pre-diagnostic study revealed that circulating AA, an N-6 polyunsaturated fatty acid derived from LA, was positively associated with the risk of ALS [13]. Due to the extremely low concentrations of polyunsaturated fatty acids in plasma and the limited sample sizes of that study, the measurement error cannot be neglected. Our study adopted the MR framework and involved large-scale genetic data, ensuring reliable statistical power. Studies have shown that LA-rich food increases oxidative stress and inflammation [54,55], both of which contribute to ALS [56]. LA enhances tumor necrosis factor (TNF)-mediated oxidative stress [57], while TNF-α, a potent inflammatory cytokine, has been found to induce apoptosis and to contribute to oxidative stress by activating microglia in an SOD1-G93A mouse model [58]. In addition, LA is considered to limit the synthesis of long-chain n-3 PUFAs from alpha-linolenic acid [59,60], while the intake of n-3 PUFA-enriched foods can prevent or delay the onset of ALS by reducing inflammation and providing neuroprotection [14,61]. Therefore, excessive LA may create a potentially inflammatory environment that increases the risk of ALS [62]. However, no relationship was found between n-3 PUFAs and ALS in our research, which may be due to the limited IV numbers of n-3 PUFAs and thereby the finite proportion of phenotypic variance explanation. In the future, more experiments are needed to explore the exact mechanism underlying LA and ALS.

Among the vitamins, we found an inversely causal association between two fat-soluble vitamins, namely, vitamin D and vitamin E, and ALS, while water-soluble vitamins showed negative results. Vitamin D is a steroid molecule that is produced in the skin by exposure to the ultraviolet spectrum of sunlight or is obtained from food and biological dietary supplements [63]. It has been reported that severe vitamin D deficiency is independently associated with significantly accelerated ALS progression and reduced life expectancy [64,65]. Since overactivation of glutamate receptors leads to continuously increased cytoplasmic calcium concentrations and subsequent ALS burden [66,67], vitamin D can exert neuroprotective effects by upregulating calcium-binding proteins and reducing glutamate-induced caspase-3 activity [68,69]. In addition, vitamin D can assist neurotrophic factors in protecting motoneurons [64]. However, contrary to our findings, Susanna C Larsson et al. reported a null causal association between serum 25OHD concentrations and ALS risk [70], which may be due to the different selections of GWAS data. The 25OHD GWAS [71] utilized in their study included 79,366 Europeans and yielded only two more new loci (SEC23A and AMDHD1) than previous 25OHD GWAS findings and explained approximately 7.5% of the heritability of 25OHD. Due to the gene prediction level of 25OHD depending on a medium polygenic structure, the statistical power of the 25OHD GWAS used was insufficient. To compensate for this shortcoming, we enrolled the 25OHD GWAS with the largest sample size. Similarly, a larger ALS GWAS and various sensitivity analyses were employed in our study to ensure convincing and robust estimates.

Vitamin E is another common essential vitamin [72], which mainly provides antioxidative protection against lipid peroxidation, reactive oxygen species (ROS), and reactive nitrogen species (RNS) [73]. The results of several large prospective cohort studies implied that the regular use of vitamin E supplements has a time-dependent protective effect on ALS [10,74]. In addition, vitamin E was suggested to have a synergistic effect with n-3 polyunsaturated fatty acids on reducing the risk of ALS [75]. In this category, we adopted two respective sets of instruments to measure the absolute and relative concentrations of vitamins, aiming to generate robust results. Although both sets for retinol and ascorbate yield similar conclusions, a significant association between the relative concentration of alpha-tocopherol and the risk of ALS was not found for absolute alpha-tocopherol. Owing to the high overlapping IVs of absolute alpha-tocopherol and lipid metabolism [76], its protective effect may be offset by the strong risk that lipids pose in ALS. Moreover, the number of IVs is too limited to develop MR–PRESSO analysis; therefore, pleiotropic bias cannot be ruled out, which violates the MR–InSIDE hypothesis. The IVs adopted for relative alpha-tocopherol have allowed for circumvention of these hurdles. Hence, our findings, in combination with previous literature, indicate that vitamin E supplementation may be a preventive agent for ALS. However, vitamin E and fat always coexist in the diet, making it necessary to comprehensively consider the complex effects of the overall diet on ALS, where using vitamin supplements may be advisable.

This pioneering study provided evidence for the primary prevention of ALS from a dietary perspective. We found that ALS-associated essential nutrients were concentrated in polyunsaturated fatty acids and fat-soluble vitamins, providing supportive evidence of the importance of lipid metabolism in ALS. No evidence has shown that diet-derived circulating essential amino acids and minerals influence the risk of ALS in the general population, even though some of them (e.g., branched-chain amino acids [77], iron, zinc, and selenium [78]) have attracted widespread attention. However, our research has certain limitations. First, since MR analyses assume linearity, potential nonlinear associations may bias the results, whereas we were unable to confirm whether the relationships between these essential nutrients and ALS risk are non-U-shaped or are other shapes. Second, some vitamins have a variety of compounds. A previous study proposed that the plasma alpha-tocopherol concentration was not an ideal biomarker for vitamin E status [79]. The survival of circulating alpha-tocopherol was longer in individuals with high lipid levels, possibly related to reduced lipoprotein catabolism and tissue absorption. As the best biomarkers for vitamin status cannot be determined via MR approaches, we still leveraged currently widely recognized active forms of vitamins as their proxies, which have mature measurement techniques and biological mechanisms, allowing for more clinically applicable conclusions than those for the unusual forms. It is tempting to select other naturally occurring vitamin forms to verify our findings. Third, our results suggested a potential influence of dietary nutrient supplementation on ALS prevention. The estimates according to genetic variance reflect the effects of lifetime accumulation exposure to a biomarker. Therefore, the short-term efficacy of nutrient supplementation for diseases should be interpreted with caution. Fourth, given that different nutrients play specific roles in biological processes, their protective effect on the overall ALS population may be driven by a strong effect on a certain subgroup of diseases. For example, vitamin D can maintain calcium homeostasis, which may be extremely beneficial for those with a severe calcium imbalance. However, the lack of individual-level ALS GWAS data makes it hard to validate this hypothesis accurately. Perhaps subsequent experiments could pay attention to this issue. Moreover, when genetic data become available for ALS patients with clinical phenotypes, future similar studies should be performed to assess whether dietary nutrients provide modifying effects on ALS progression and not just consider disease prevention.

## 5. Conclusions

Evidence from our study demonstrated that vitamin D and vitamin E are protective against the risk of ALS, and LA is suggested to be positively associated with ALS risk. It is unlikely that a clinical benefit for the primary prevention of ALS will be achieved through supplementing essential amino acids and minerals. Our study classified and profiled the effects of specific diet-derived circulating nutrients on the risk of ALS, providing evidence for medical interventions for ALS in the general population from the perspective of gene–environmental interactions.

## Figures and Tables

**Figure 1 nutrients-14-00920-f001:**
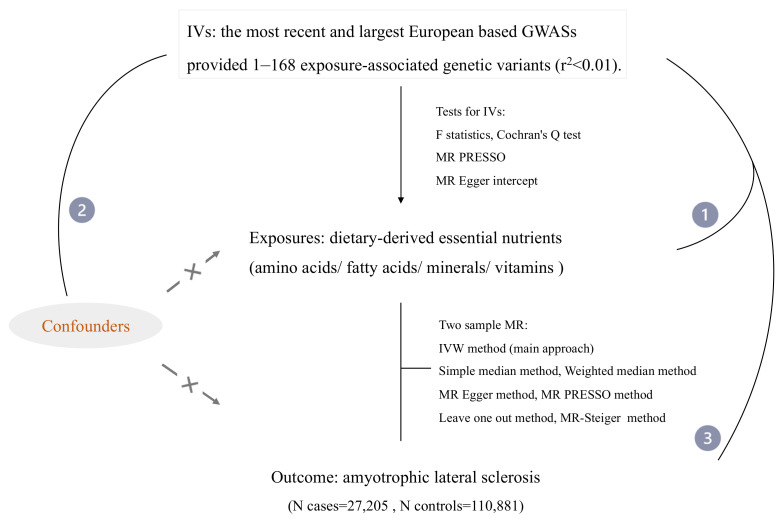
Flow chart showing the process for the Mendelian randomization analyses. The number in the line indicates 3 key assumptions for MR. Assumption 1: The selected genetic variants are not related to other confounders. Assumption 2: The selected genetic variants are significantly related to the exposure. Assumption 3: The selected genetic variants are significantly related to the risk of the outcome only through the pathway from exposure. IVs, instrumental variables; GWASs, genome-wide association studies; MR, Mendelian randomization; IVW, inverse variance–weighted; MR–PRESSO, Mendelian randomization pleiotropy residual sum and outlier.

**Figure 2 nutrients-14-00920-f002:**
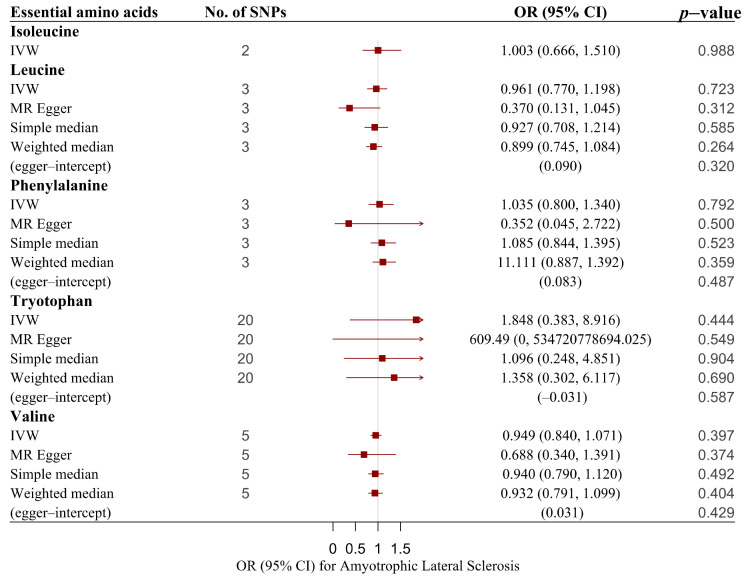
Associations of genetically predicted essential amino acids with risk of ALS using main and sensitivity MR analyses. ALS, amyotrophic lateral sclerosis; IVW, inverse variance–weighted; MR, Mendelian randomization; OR, odds ratio; 95% CI, 95% confidence interval; SNPs, single nucleotide polymorphisms.

**Figure 3 nutrients-14-00920-f003:**
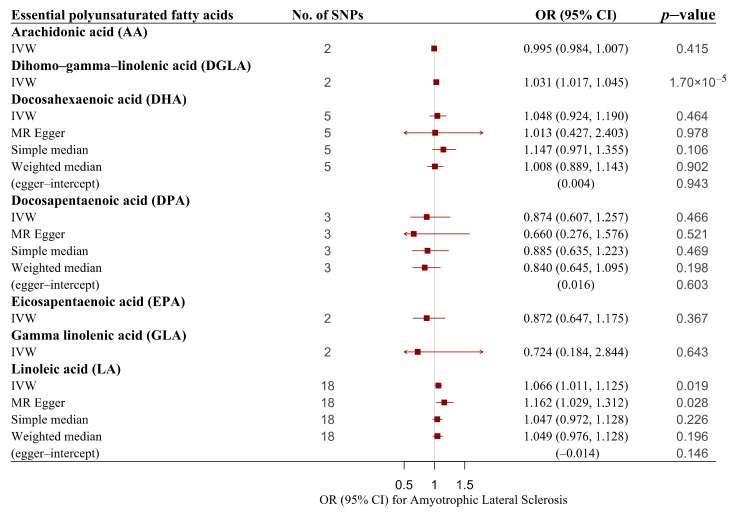
Associations of genetically predicted essential polyunsaturated fatty acids with risk of ALS using main and sensitivity MR analyses. ALS, amyotrophic lateral sclerosis; IVW, inverse variance–weighted; MR, Mendelian randomization; OR, odds ratio; 95% CI, 95% confidence interval; SNPs, single nucleotide polymorphisms.

**Figure 4 nutrients-14-00920-f004:**
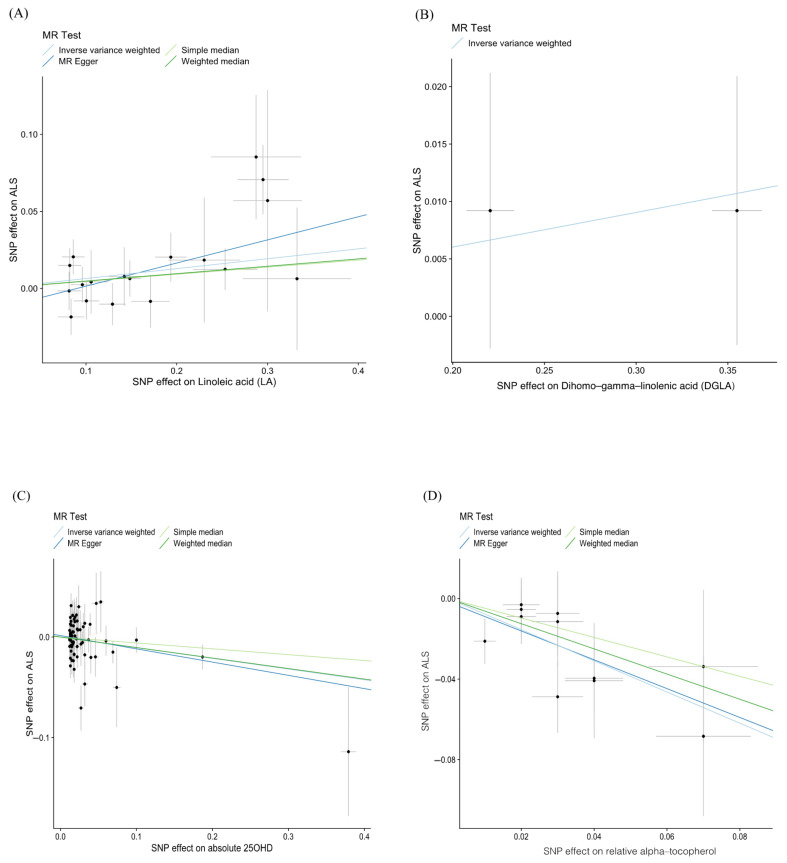
Scattered plots of the casual effect of essential nutrients on ALS. The x-axis plots the previously published β–estimate for the association between each SNP and essential nutrients. The y-axis plots the β–estimate for the association between each SNP and risk of ALS. The slope of each line corresponds to the estimated MR effect per method. The examined essential nutrients included (**A**) linoleic acid (LA), (**B**) Dihomo–gamma–linoleic acid (DGLA), (**C**) absolute 25OHD, and (**D**) relative alpha-tocopherol. ALS, amyotrophic lateral sclerosis; IVW, inverse variance–weighted; MR, Mendelian randomization; OR, odds ratio; 95% CI, 95% confidence interval; SNPs, single nucleotide polymorphisms.

**Figure 5 nutrients-14-00920-f005:**
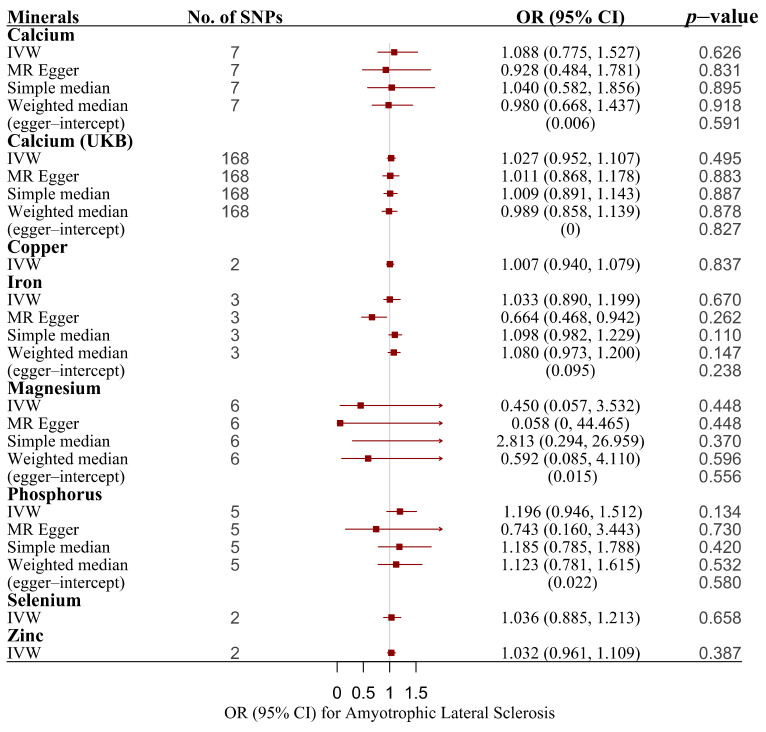
Associations of genetically predicted essential minerals with risk of ALS using main and sensitivity MR analyses. ALS, amyotrophic lateral sclerosis; IVW, inverse variance–weighted; MR, Mendelian randomization; OR, odds ratio; 95% CI, 95% confidence interval; SNPs, single nucleotide polymorphisms.

**Figure 6 nutrients-14-00920-f006:**
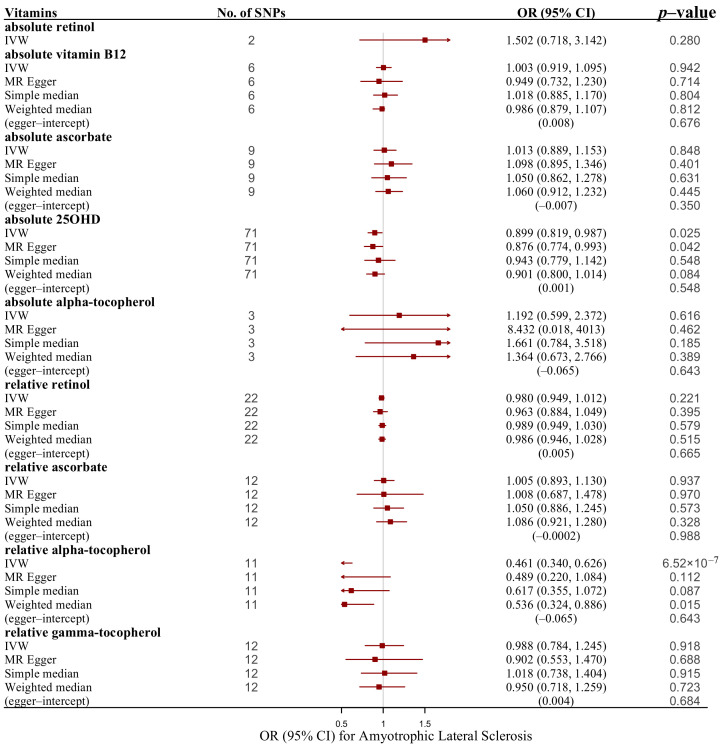
Associations of genetically predicted essential vitamins with risk of ALS using main and sensitivity MR analyses. ALS, amyotrophic lateral sclerosis; IVW, inverse variance–weighted; MR, Mendelian randomization; OR, odds ratio; 95% CI, 95% confidence interval; SNPs, single nucleotide polymorphisms.

**Table 1 nutrients-14-00920-t001:** Summary of the causal effects of each trait on ALS via different MR methods.

Exposures	Methods	No. SNPs	OR (95%CI)	*p* Value
Essential amino acids				
Isoleucine	IVW	2	1.003 (0.666, 1.510)	0.988
Leucine	IVW	3	0.961 (0.770, 1.198)	0.723
Leucine	MR–Egger	3	0.370 (0.131, 1.045)	0.312
Leucine	Simple median	3	0.927 (0.708, 1.214)	0.583
Leucine	Weighted median	3	0.899 (0.745, 1.084)	0.264
Phenylalanine	IVW	3	1.035 (0.800, 1.340)	0.792
Phenylalanine	MR–Egger	3	0.352 (0.045, 2.722)	0.500
Phenylalanine	Simple median	3	1.085 (0.844, 1.395)	0.523
Phenylalanine	Weighted median	3	1.111 (0.887, 1.392)	0.359
Tryptophan	IVW	20	1.848 (0.383, 8.916)	0.444
Tryptophan	MR–Egger	20	609.49 (0, 534, 720, 778, 694.025)	0.549
Tryptophan	Simple median	20	1.096 (0.248, 4.851)	0.904
Tryptophan	Weighted median	20	1.358 (0.302, 6.117)	0.690
Valine	IVW	5	0.949 (0.840, 1.071)	0.397
Valine	MR–Egger	5	0.688 (0.340, 1.391)	0.374
Valine	Simple median	5	0.94 (0.790, 1.120)	0.492
Valine	Weighted median	5	0.932 (0.791, 1.099)	0.404
Essential polyunsaturated fatty acids				
Arachidonic acid (AA)	IVW	2	0.995 (0.984, 1.007)	0.415
Dihomo-gamma-linolenic acid (DGLA)	IVW	2	1.031 (1.017, 1.045)	1.70 × 10^−5^
Docosahexaenoic acid (DHA)	IVW	5	1.048 (0.924, 1.190)	0.464
Docosahexaenoic acid (DHA)	MR–Egger	5	1.013 (0.427, 2.403)	0.978
Docosahexaenoic acid (DHA)	Simple median	5	1.147 (0.971, 1.355)	0.106
Docosahexaenoic acid (DHA)	Weighted median	5	1.008 (0.889, 1.143)	0.902
Docosapentaenoic acid (DPA)	IVW	3	0.874 (0.607, 1.257)	0.466
Docosapentaenoic acid (DPA)	MR–Egger	3	0.660 (0.276, 1.576)	0.521
Docosapentaenoic acid (DPA)	Simple median	3	0.885 (0.635, 1.233)	0.469
Docosapentaenoic acid (DPA)	Weighted median	3	0.840 (0.645, 1.095)	0.198
Eicosapentaenoic acid (EPA)	IVW	2	0.872 (0.647, 1.175)	0.367
Gamma linolenic acid (GLA)	IVW	2	0.724 (0.184, 2.844)	0.643
Linoleic acid (LA)	IVW	18	1.066 (1.011, 1.125)	0.019
Linoleic acid (LA)	MR–Egger	18	1.162 (1.029, 1.312)	0.028
Linoleic acid (LA)	Simple median	18	1.047 (0.972, 1.128)	0.226
Linoleic acid (LA)	Weighted median	18	1.049 (0.976, 1.128)	0.196
Minerals				
Calcium	IVW	7	1.088 (0.775, 1.527)	0.626
Calcium	MR–Egger	7	0.928 (0.484, 1.781)	0.831
Calcium	Simple median	7	1.04 (0.582, 1.856)	0.895
Calcium	Weighted median	7	0.98 (0.668, 1.437)	0.918
Calcium (UKB)	IVW	168	1.027 (0.952, 1.107)	0.495
Calcium (UKB)	MR–Egger	168	1.011 (0.868, 1.178)	0.883
Calcium (UKB)	Simple median	168	1.009 (0.891, 1.143)	0.887
Calcium (UKB)	Weighted median	168	0.989 (0.858, 1.139)	0.878
Copper	IVW	2	1.007 (0.940, 1.079)	0.837
Iron	IVW	3	1.033 (0.890, 1.199)	0.670
Iron	MR–Egger	3	0.664 (0.468, 0.942)	0.262
Iron	Simple median	3	1.098 (0.982, 1.229)	0.100
Iron	Weighted median	3	1.080 (0.973, 1.200)	0.147
Magnesium	IVW	6	0.450 (0.057, 3.532)	0.448
Magnesium	MR–Egger	6	0.058 (0, 44.465)	0.448
Magnesium	Simple median	6	2.813 (0.294, 26.959)	0.370
Magnesium	Weighted median	6	0.592 (0.085, 4.110)	0.596
Phosphorus	IVW	5	1.196 (0.946, 1.512)	0.134
Phosphorus	MR–Egger	5	0.743 (0.160, 3.443)	0.730
Phosphorus	Simple median	5	1.185 (0.785, 1.788)	0.420
Phosphorus	Weighted median	5	1.123 (0.781, 1.615)	0.532
Selenium	IVW	2	1.036 (0.885, 1.213)	0.658
Zinc	IVW	2	1.032 (0.961, 1.109)	0.387
Vitamins				
absolute retinol	IVW	2	1.502 (0.718, 3.142)	0.280
absolute vitamin B-12	IVW	6	1.003 (0.919, 1.095)	0.942
absolute vitamin B-12	MR–Egger	6	0.949 (0.732, 1.230)	0.714
absolute vitamin B-12	Simple median	6	1.018 (0.885, 1.170)	0.804
absolute vitamin B-12	Weighted median	6	0.986 (0.879, 1.107)	0.812
absolute ascorbate	IVW	9	1.013 (0.889, 1.153)	0.848
absolute ascorbate	MR–Egger	9	1.098 (0.895, 1.346)	0.401
absolute ascorbate	Simple median	9	1.050 (0.862, 1.278)	0.631
absolute ascorbate	Weighted median	9	1.060 (0.912, 1.232)	0.445
absolute 25OHD	IVW	71	0.899 (0.819, 0.987)	0.025
absolute 25OHD	MR–Egger	71	0.876 (0.774, 0.993)	0.042
absolute 25OHD	Simple median	71	0.943 (0.779, 1.142)	0.548
absolute 25OHD	Weighted median	71	0.901 (0.800, 1.014)	0.084
absolute alpha-tocopherol	IVW	3	1.192 (0.599, 2.372)	0.616
absolute alpha-tocopherol	MR–Egger	3	8.432 (0.018, 4012.598)	0.621
absolute alpha-tocopherol	Simple median	3	1.661 (0.784, 3.518)	0.185
absolute alpha-tocopherol	Weighted median	3	1.364 (0.673, 2.766)	0.389
relative retinol	IVW	22	0.980 (0.949, 1.012)	0.221
relative retinol	MR–Egger	22	0.963 (0.884, 1.049)	0.395
relative retinol	Simple median	22	0.989 (0.949, 1.030)	0.579
relative retinol	Weighted median	22	0.986 (0.946, 1.028)	0.515
relative ascorbate	IVW	12	1.005 (0.893, 1.130)	0.937
relative ascorbate	MR–Egger	12	1.008 (0.687, 1.478)	0.970
relative ascorbate	Simple median	12	1.050 (0.886, 1.245)	0.573
relative ascorbate	Weighted median	12	1.086 (0.921, 1.280)	0.328
relative alpha-tocopherol	IVW	11	0.461 (0.340, 0.626)	6.52 × 10^−7^
relative alpha-tocopherol	MR–Egger	11	0.489 (0.220, 1.084)	0.112
relative alpha-tocopherol	Simple median	11	0.617 (0.355, 1.072)	0.087
relative alpha-tocopherol	Weighted median	11	0.536 (0.324, 0.886)	0.015
relative gamma-tocopherol	IVW	12	0.988 (0.784, 1.245)	0.918
relative gamma-tocopherol	MR–Egger	12	0.902 (0.553, 1.470)	0.688
relative gamma-tocopherol	Simple median	12	1.018 (0.738, 1.404)	0.915
relative gamma-tocopherol	Weighted median	12	0.950 (0.718, 1.259)	0.723

**Table 2 nutrients-14-00920-t002:** Summary of the additional MR analysis for the effect of each non ALS-associated trait on ALS; * the F statistics are the mean F value for each eligible instrumental variable.

Exposures	F Statistics *	R2	MR–Egger		Cochran’s Q		MR–PRESSO				MR–Steiger	
			Intercept	*p*	Q	*p*	RS Sobs	P- Global Test	P- Outlier	P- Distortion	Correct Causal Direction	*p*
Essential amino acids												
Isoleucine	32.07	0.003	NA	NA	4.333	0.037	NA	NA	NA	NA	TRUE	4.74 × 10^−23^
Leucine	32.532	0.004	0.09	0.32	4.049	0.132	NA	NA	NA	NA	TRUE	3.27 × 10^−38^
Phenylalanine	26.032	0.004	0.083	0.487	3.901	0.142	NA	NA	NA	NA	TRUE	7.23 × 10^−31^
Tryptophan	16.831	0.043	–0.031	0.587	52.831	0	58.666	<2 × 10^−4^	0.893	0.074	TRUE	4.08 × 10^−134^
Valine	29.951	0.006	0.031	0.429	3.423	0.49	4.651	0.587	NA	NA	TRUE	7.73 × 10^−61^
Polyunsaturated fatty acids												
Arachidonic acid (AA)	1302.627	0.233	NA	NA	0.725	0.395	NA	NA	NA	NA	TRUE	2.57 × 10^−198^
Docosahexaenoic acid (DHA)	19.211	0.007	0.004	0.943	5.774	0.217	9.826	0.26	NA	NA	TRUE	2.22 × 10^−49^
Docosapentaenoic acid (DPA)	129.675	0.044	0.016	0.603	3.878	0.144	NA	NA	NA	NA	TRUE	5.75 × 10^−192^
Eicosapentaenoic acid (EPA)	68.305	0.016	NA	NA	1.321	0.25	NA	NA	NA	NA	TRUE	8.54 × 10^−63^
Gamma linolenic acid (GLA)	77.298	0.018	NA	NA	0.993	0.319	NA	NA	NA	NA	TRUE	8.70 × 10^−76^
Minerals												
Calcium	22.182	0.003	0.006	0.591	6.118	0.41	8.54	0.486	NA	NA	TRUE	7.48 × 10^−85^
Calcium (UKB)	25.391	0.014	0	0.827	186.865	0.139	188.91	0.152	NA	NA	TRUE	0
Copper	18.08	0.014	NA	NA	0.835	0.361	NA	NA	NA	NA	TRUE	3.71 × 10^−28^
Iron	159.741	0.01	0.095	0.238	6.747	0.034	NA	NA	NA	NA	TRUE	9.72 × 10^−127^
Magnesium	20.988	0.005	0.015	0.556	11.06	0.05	19.824	0.1	NA	NA	TRUE	4.97 × 10^−73^
Phosphorus	18.29	0.004	0.022	0.58	2.733	0.603	4.43	0.633	NA	NA	TRUE	2.82 × 10^−50^
Selenium	39.218	0.008	NA	NA	2.291	0.13	NA	NA	NA	NA	TRUE	8.59 × 10^−42^
Zinc	23.67	0.018	NA	NA	0.966	0.326	NA	NA	NA	NA	TRUE	4.77 × 10^−28^
Vitamins												
absolute retinol	24.943	0.01	NA	NA	2	0.157	NA	NA	NA	NA	TRUE	7.89 × 10^−25^
absolute vitamin B-12	75.036	0.012	0.008	0.676	5.109	0.403	0.774	0.366	NA	NA	TRUE	3.62 × 10^−212^
absolute ascorbate	20.303	0.004	−0.007	0.35	8.331	0.402	9.643	0.506	NA	NA	TRUE	9.77 × 10^−120^
absolute alpha-tocopherol	3.121	0.001	−0.065	0.643	4.022	0.134	NA	NA	NA	NA	TRUE	1.04 × 10^−24^
relative retinol	4.158	0.047	0.005	0.665	28.945	0.115	31.643	0.13	NA	NA	TRUE	1.59 × 10^−118^
relative ascorbate	83.86	0.04	−0.0002	0.988	9.901	0.539	11.769	0.55	NA	NA	TRUE	1.17 × 10^−62^
relative gamma-tocopherol	5.881	0.012	0.004	0.684	14.503	0.206	17.795	0.214	NA	NA	TRUE	2.71 × 10^−52^

NA, Due to the limitation of the number of SNPs, no valid value can be obtained. True, The assumed causal relationship is in the right direction.

**Table 3 nutrients-14-00920-t003:** Summary of the additional analysis for the causal effects of each ALS-associated trait on ALS.

Methods		Dihomo-Gamma-Linolenic Acid (DGLA)	Linoleic Acid (LA)	Absolute 25-OHD	Relative Alpha-Tocopherol
mean F statistics		218.419	17.009	89.901	4.07
R2		0.049	0.023	0.014	0.006
MR–Egger	intercept	NA	−0.014	0.001	−0.065
	*p* value	NA	0.146	0.548	0.643
Cochran’s Q	Q	0.062	20.419	84.75	6.652
	*p* value	0.804	0.253	0.11	0.758
MR–PRESSO	RS Sobs	NA	23.102	85.808	7.907
	P-global test	NA	0.258	0.144	0.807
	P-outlier	NA	NA	NA	NA
	P-distortion	NA	NA	NA	NA
MR–Steiger	correct causal direction	TRUE	TRUE	TRUE	TRUE
	*p* value	4.37 × 10^−207^	2.11 × 10^−207^	0	1.98 × 10^−45^

NA, Due to the limitation of the number of SNPs, no valid value can be obtained. True, The assumed causal relationship is in the right direction.

## Data Availability

All data generated or analyzed during this study are included in this published article and its Appendix A. Codes generated or used during the study are available from the corresponding author by request.

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
