# Peer review of "Dietary-Derived Essential Nutrients and Amyotrophic Lateral Sclerosis: A Two-Sample Mendelian Randomization Study"

_nutrients, 2022, doi:10.3390/nu14050920_

Round 1
Reviewer 1 Report
It is very important manuscript with a solid conclusion about the positive impact of vitamin D and vitamin E as protectives against the risk of ALS. Also, LA is suggested to be positively associated with ALS risk.
Line 201: Delete "Figure 1" (appears twice)
Improve the presentation of Figure 2 and 3. They are not sharp.
Reviewer 2 Report
It seems that the manuscript is well structured and analyzes important factors on the subject of reference. There is a sufficient number of updated references. The design applied in the study (Mendelian randomization) is adequate to examine in observational studies the causal effect that a modifiable risk factor, such as diet, can have on a disease. However, in table 3 there is a value that appears to be in error, specifically in the triotophane method by MR Egger. Despite this, the conclusions are consistent with the results and the published literature.